# Recycling of EPDM via Continuous Thermo-Mechanical Devulcanization with Co-Rotating Twin-Screw Extruder

**DOI:** 10.3390/polym14224853

**Published:** 2022-11-11

**Authors:** Valentina Brunella, Veronica Aresti, Umberto Romagnolli, Bruno Muscato, Marco Girotto, Paola Rizzi, Maria Paola Luda

**Affiliations:** 1Dipartimento di Chimica, Università di Torino, Via P. Giuria 7, 10125 Turin, Italy; 2F.lli Maris S.p.A., Corso Moncenisio 22, 10090 Rosta, Italy

**Keywords:** recycling, devulcanization, EPDM, co-rotating twin-screw-extruder, waste valorization

## Abstract

Devulcanization represents the recycling of choice for a homogenous rubber waste stream because it allows revulcanization of samples previously devulcanized, making the life of the rubber virtually endless, according to the principles of circular economy. Among the many devulcanization processes, the thermo-mechanical one is the most appealing because it is a continuous process, easy to be industrialized. In this paper a comprehensive set of analyses (FTIR, TGA, DSC, elemental analyses, Py-GC/MS, swelling tests) were carried out on a post-industrial ethylene propylene diene monomer (EPDM), thermo-mechanical devulcanized in a co-rotating twin-screw extruder with different process parameters (thermal and screw profile, rpm). Results of the swelling test according to the Flory–Rehner theory and Horikx analyses show that the higher the thermal profile and the higher the rpm, the higher is the percentage of devulcanization. The quality of the devulcanized sample in terms of sol fraction and percentage of random scissions depends on the process conditions. The screw profile concurs to the efficiency of the devulcanization: the different number of kneading elements and more in general the screw profile composition affects the percentage of devulcanization, making the results in some tests more dependent on the screw speed.

## 1. Introduction

Elastomers deform instantly under load application and soon return to their original size by load remotion. The structural reason for elastomeric behavior is the presence of flexible chains connected in a network with chemical unreversible crosslinks in the thermosetting elastomers or physical reversible crosslinks in the thermoplastic elastomers. Due to the permanent crosslinking, thermosetting elastomers are difficult to be recycled.

Ethylene propylene diene monomer (EPDM) is a synthetic thermosetting elastomer. It is a copolymer of propylene and ethylene combined with pendant diene to crosslink the polymer. It shows strong resistance to many external agents such as heat, ozone, mild acids, synthetic brake fluids, water, ethylene glycol, and other liquids. Despite being an inert material with limited environmental impact, EPDM is used in a chemically cross-linked form in a variety of applications and this feature makes its recycling difficult. EPDM rubber is mainly vulcanized by two methods: (a) with sulfur, creating sulfuric crosslinks; or (b) with peroxides, linking polymer chains through C–C covalent bonds [1]. The market for ethylene propylene diene monomer was valued at over 1600 kilotons in 2021 and the compound annual growth rate (CAGR) is expected to be more than 4% during the period 2022–2027. This relevant global demand for EPDM accounts for the massive expansion of the application sectors including automotive, building and construction, and domestic appliances [2,3]. Therefore, an efficient recycling process for EPDM rubber can effectively reduce the environmental impacts of this material and can promote an efficient circular economy.

It should be emphasized that the EPDM parts in most goods (such as seals in domestic appliances and car doors/windscreens) can be relatively easily dismantled and collected separately, constituting an important homogeneous waste stream which can be valorized according to the concepts of the circular economy [4]. Therefore, specific recycling processes must be used, such as those involving de-crosslinking.

Generally, rubber recycling can be carried out in three strategies: fragmentation, pyrolysis, devulcanization or reclaiming.

Fragmentation, the most widely traditional approach for rubber recycling, is carried out by grinding the rubber to powdered particles that can be blended with thermoplastics or mixed with virgin EPDM compounds as a filler, to extend their life. A variety of products can be obtained in this way (thermoplastic elastomers, playground and road surfaces, etc.). This is an example of linear economy where the scraps progressively decrease their value; in addition, the market for these products is limited, hence the quality of recycled rubber products must be improved especially in the perspective of the circular economy [5].

Pyrolysis is a sort of tertiary feedstock recycling in which feeding is converted merely by thermal energy into products such as monomers or pyrolysis oil depending on the polymer fed to the reactor. It does not require preliminary separation of the different polymers and is particularly suitable for mixed heterogeneous flow of waste or complex multicomponent items [6].

Conventional reclamation processes (Figure 1a) involve the use of chemicals, high temperature, and high pressure to cleave the crosslink in an unrestricted way leading to a highly degraded material with poor properties due to severe main chain degradation. The processes applied do not discriminate main chain and crosslink cleavage [7].

Devulcanization processes (Figure 1b) cleave the crosslinks with no or small main chain scission. The basic idea of devulcanization is to break down the monosulfide (C–S), disulfide (S–S), and polysulfide bonds formed during vulcanization of EPDM enabling the reclaimed EPDM chains to be revulcanized, resulting in an ideally endless life [8].

Among the four methods mentioned, devulcanization is the most appropriate process for a homogeneous waste stream because it better fits the circular recycling concepts, rubber scraps being converted into a material that can be revulcanized [9,10].

A few technologies of rubber devulcanization can be performed; for instance, chemical techniques use a devulcanizing agent to breakdown the crosslinked network. Among them, organic disulfides, such as 2,2′-dibenzothiazole disulfide, and dixylyl disulfide are the most commonly used to break down the crosslinked network. The devulcanization is carried out as a batch process by refluxing a mixture of rubber with the appropriate solvent and reclaiming agents [11,12].

On the contrary, physical techniques use physical agents such as heat, mechanical stress, microwaves, or ultrasound, alone or in combination, to break down the crosslinking bonds [13,14,15,16].

Because of the weak penetration of devulcanization agents in the rubber matrix, mechanical devulcanization is a more promising route than chemical devulcanization [17].

Thermo-mechanical devulcanization by extrusion seems to be even more promising, being a continuous process, easy to be industrialized. Usually, several parameters are needed to fully describe this reactive extrusion process and possibly to model it considering the experimental input data [18]. In the thermomechanical method, the materials are exposed to high shear stress at a given temperature [19]. In addition, being a continuous process easy to be industrialized, thermo-mechanical devulcanization is of special interest for rubbers that can be collected separately as in the case of EPDM, a rubber usually difficult to recycle via conventional chemical devulcanization techniques. EPDM from special waste streams, such as that from dismantling of white equipment in WEEE (waste of electrical and electronic equipment) or end of life vehicles.

One of the first mechanisms for thermo-mechanical devulcanization under shear was formulated by Mouri et al. [20,21].

The strength of the carbon bonds –C–C– of the rubber chain segments, monosulfidic bonds –S–, disulfidic bonds –S–S– and polysulfidic bonds –S–S–S─ are 352, 285, 268, and 251 kJ/mol [22]. In the case of sulfur-crosslinked rubbers, the heat converts the polysulfidic (PS) and disulfidic (DS) crosslinks into monosulfidic (MS) crosslinks. Furthermore, the monosulphide bond is broken by addition of shear stress and finally the devulcanized rubber is obtained. Recently the contribution of 3D network and defects have been considered to describe the behavior of vulcanized materials [23].

Another model for selective network scission by thermo-mechanical devulcanization is based on different elastic constants and binding energies of S–S, C–S, and C–C bonds. The differences between the binding energies are minor, and a purely thermal process leads to unselective cleavage. Since the elastic constant of S–S bonds is much smaller than that of C–C bonds, S–S bonds are the most extended under shear. As a result, the elastic energy generated by shear stress is most significant at S–S bonds leading to selective cleavage of crosslinks. However, this cleavage occurs only at high shear stress, otherwise, the entropic effect predominates [24].

As far as thermo-mechanical devulcanization is concerned, apart from the undoubted advantage of being a continuous process easy to be industrialized, life cycle analysis (LCA) evidenced that the environmental impacts associated with this recycling process are limited, with the only concern being related to energy consumption [25]. Therefore, the use of the thermo-mechanical devulcanization process can help in limiting the environmental burdens of the EPDM production and the adoption of more appropriate conditions either side environment or side efficiency are required.

## 2. Materials and Methods

Post-industrial EPDM seals came from the portholes of washing machines. The EPDM seals were ground by F.lli Maris S.p.A. obtaining a free-flowing form material ready for feeding into the extruder. The devulcanized rubber, for each test, was collected and cut into pieces (approximatively 0.5 × 0.5 × 0.5 cm) for sample characterization and swelling tests.

Devulcanization process: the devulcanization process using a co-rotating twin-screw extruder without the use of solvents/devulcanizing agents was studied, designed, and patented by F.lli Maris S.p.A.—an extruder manufacturer—in 2011.

The co-rotating laboratory twin-screw extruder used to carry out the current study has the following technical characteristics: screw diameter = 20 mm, extruder length 64 L/D, which corresponds to 16 barrels (each barrel length is four times the diameter of the screw). Each barrel is equipped with a heating and a water-cooling system to guarantee precise control on the barrel temperature of the most suitable process temperature. The extruder was also equipped with water injection, a side degassing unit, and a sheet extrusion die. Figure 1 shows a typical layout of the Maris devulcanization line (EVOREC RUBBER) together with the main equipment.

To devulcanize the EPDM rubber and subsequently to optimize the process parameters to increase the quality of the devulcanized product, the following parameters were changed: thermal profile (high, medium, and low), screw speed (high, medium, low, and very low rpm), screw profile configuration (three different screw profiles), and water injection (0%, 10%, and 15%, where this percentage value refers to the rubber output).

The screw profiles were designed to heat and to process the rubber in order to transform it—inside the extruder—from a vulcanized solid state to a plastomeric devulcanized state. Depending on the process parameters and the combination of them, the rubber is subjected to thermal energy and mechanical energy in different ratios, but only thanks to an appropriate balancing of all the parameters is it possible to reach the right temperature of the rubber and make devulcanization possible. If these parameters are not balanced correctly, however, the rubber exiting the extruder will be pulverized and degraded. The barrel temperatures were set between 200 and 300 °C.

The injection of water in the final phase of the extrusion (devulcanization) process aims to rapidly cool the devulcanized rubber inside the extruder preserving it from possible degradation and it is also an excellent stripping agent for the reduction of gases developed during the process. In fact, the water injected into the extruder, after its mixing with the devulcanized rubber, is removed by means of the degassing unit.

The same output was used for all tests (3.5 kg/h); Table 1 shows the process conditions of each test.

**Infrared analysis.** The FTIR spectra were registered with a Perkin Elmer FTIR Spectrum 100, in ATR modality (diamond crystal with ZnSe layer). The internal surface of samples was analyzed. The qualitative composition of the samples was determined.

**Pyrolysis–GC/MS**. A Frontier Lab pyrolizer connected to a GC/MS Agilent 6890 Series and Agilent 5973 Network was used for analysis. The pyrolysis temperature was set to 600 °C, for 12 s.

**TGA**. A TA Q500 (Waters, TA thermal analysis) equipment was used to ascertain the quantitative sample composition in terms of volatiles, polymer, CaCO_3_, carbon black, and inorganic residue. The following heating program was used: nitrogen flow 60 cc/min: ramp 10 °C/min from 50 °C to 300 °C, isothermal for 10 min; ramp 10 °C/min to 550 °C. Switched to air 60 cc/min: isothermal for 10 min, ramp 20 °C/min to 650 °C, isothermal for 15 min; ramp 20 °C/min to 800 °C, isothermal for 2 min.

Under these conditions plasticizing oils evaporate along the isothermal segment at 300 °C, rubber degrades in the first ramp in nitrogen up to 550 °C, carbon black, if present, is oxidized to CO_2_ in the plateau at 550 °C in air and CaCO_3_ decomposes to CO_2_ (volatile) and CaO (in the residue) in the following ramp.

**DSC**. A DSC Q200 (Waters, TA thermal analysis) equipment was used to ascertain the glass transition temperature of the samples either in the original, or acetone extracted, or after-SW series as described below (swelling test paragraph). The following heating program was used: Heat cool heat method: equilibrate at −85 °C, ramp 20 °C/min to 80 °C (first heating). Ramp 20 °C/min to 85 °C (cooling). Ramp 20 °C/min to 80 °C (second heating). Tg was taken at the inflection point of the second heating. Duplicate analyses were carried out, the error was ±0.5 °C.

**Elemental analysis.** Elemental analyses were performed with a CHNS-O Analyzer Flash EA, 1112 Series equipment on 2.5 mg of sample using V_2_O_5_ as a catalyzer, in duplicate. Error was ±0.05%

**Swelling test.** The original rubber samples (ORIGINAL), directly obtained from Maris S.p.A., were weighed and reduced to smaller pieces. They were extracted in hot acetone for 16 h using a Soxhlet apparatus to eliminate the acetone-soluble species. The solids were dried at room temperature under a laboratory fume hood until constant weight. This operation generates a second series of samples (EXTRACTED) in which plasticizing oils are removed.

The Extracted samples were weighed (*W_b_*) and immersed in 100 mL of toluene at room temperature for 24 h. After this time, the samples were quickly dried with absorbent paper, to remove the excess liquid from its surface, and immediately weighed (*W_a_*) in a cap-closed, tared, and weighted jar. Then the cap was removed from the jar and both the cap and the sample in the open jar were put into a forced-ventilating oven at 55 °C for 24 h.

The jar containing the dried sample and the cap were removed from the oven; the jar was recapped and allowed to cool at room temperature. Finally, the closed jar was weighed and the weight of dried sample calculated (*W_d_*). The amount of the absorbed solvent is *W_a_ − W_d_*, the amount of solubilized polymer is *W_b_ − W_d_* [26].

Therefore, the percentage of extracted polymer *(%sol*) is:(1)$$\%sol=\frac{W_{b}-W_{d}}{W_{b}*\phi }*100$$
*ϕ* is the percent of polymer in the acetone-extracted sample as derived from TGA.

Swelling tests generated a third series of samples (after-SW) in which the sol fraction of the polymer was removed.

The higher the amount of absorbed solvent, the lower is the crosslinking degree. From this consideration Flory–Rehner (FR) equated the free mixing energy and the elastic recovery force at the equilibrium:(2)$$\Delta G_{mix}+\Delta G_{elas}=0$$

By substituting the appropriate thermodynamic equations, the classical FR equation is recovered [27].

Their model was later extended considering the more realistic phantom model instead of the classical one of affine deformation [28].

The final equation for density of crosslinking (*n_SW_)* is:(3)$$n_{sw}=\frac{ln\left(1-\varphi_{r}\right)+\varphi_{r}+\chi \varphi_{r}^{2}}{\rho_{r}V_{s}\left(1-\frac{2}{f}\right)\varphi_{r}^{1/3}}$$

*n_SW_* (density of crosslinking) corresponds to *1/M_sub_,* with *M_sub_* the average number molecular weight of the chains between two adjacent crosslinking;*V_s_* is the molar volume of the solvent, taken from literature (here 106.52 cm^3^/mol);*f* is the functionality of crosslinking (here 4 was considered);*χ* is the interaction parameter polymer-solvent; values for *χ*_(tol,EPDM)_ ranging from 0.45 to 0.51 are reported in the literature [29] and in this paper 0.5 was used. However, variation of this value had only a minor effect on the value of the molecular weight between two crosslinks *M_sub_*;*ϕ_r_* is the volume fraction of the expander rubber and is calculated from the swelling test:


(4)
$$\varphi_{r}=\frac{W_{b}\phi \frac{1-\alpha }{\rho_{r}}}{W_{b}\phi \frac{1-\alpha }{\rho_{r}}+\frac{W_{a}-W_{d}}{\rho_{s}}}$$


*W_b_*, *W_a_,* and *W_d_* have been previously described and *ρ_r_* and *ρ_s_* are the rubber and the solvent densities, taken from literature and respectively 0.860 and 0.865 gr/cm^3^;*Φ* is the percentage of the polymer in the sample undergoing swelling, evaluated from TGA of the after-extraction sample series;*α* is the fraction of sample extracted during the swelling test.

The percentage of devulcanization of each sample *(% DVZ)_i_* is given by the following equation, which takes into account the density of crosslinking of RM (feedstock) and of each devulcanized sample:(5)$$\%DVZ_{i}=\left(1-\frac{{\left(n_{SW,\;dev}\right)}_{i}}{n_{SW,\;RM}}\right)*100$$

**Horikx analysis**. The fraction of soluble material in the degraded network is a measure of the number of scissions occurring in the network, provided that the original molecular weight of the chain before crosslinking and the average number of crosslinking/chains are known.

Under the assumption that *M* is the molecular weight of the original chains, *γ_i_*, the number of crosslinking/original chains, is:(6)$$\gamma_{i}=\frac{M}{{\left(M_{sub}\right)}_{i}}$$
and the sol fraction *S_i_* is, according to Charlesby [30]:(7)$$S_{i}=\frac{\left(2+\gamma_{i}\right)-\sqrt{\gamma_{i}^{2}+4\gamma_{i}}}{2\gamma_{i}}$$

Horikx recovered a simple relation between the soluble fraction of a network that has undergone scission and the effective number of chains in the gel fraction (as determined by swelling measurements) for two limit cases [31,32]:Chain scission at random: The required number of chain scissions is randomly applied to the not vulcanized polymer, after which the crosslinks are brought into the same positions that they would have occupied if no scission had occurred. In this case the fraction of broken links (*X_random_*) is:
(8)Xrandom=1−(1−sf12)2(1−si12)2
(subscript *f* refers to the soluble fraction of the devulcanized sample, subscript *i* to the soluble fraction of the undevulcanized crosslinked sample);

2.Severance of the crosslinks: there is no chain scission, but only opening of the crosslinks. In this case the fraction of broken links (*X_cross_*) is:


(9)
Xcross=1−γf(1−sf12)2γi(1−si12)2


In real devulcanization both random and crosslinking scissions can occur and if the sol fraction is *%sol*, the *%random* scissions is given by:(10)$${\left(\%random\right)}_{i}=\frac{{\left(\%sol\right)}_{i}-100*{\left(X_{cross}\right)}_{i}}{100*{\left(X_{random}\right)}_{i}}$$

Recently an analytical and modular model framework was put forward enabling the prediction of long-term durability, starting from fundamental principles at the molecular level and explicitly accounting for bond rupture events [33].

## 3. Results and Discussion

### 3.1. Qualitative Sample Composition

#### 3.1.1. FT-IR

FTIR spectra of RM EPDM, either original or extracted or after-SW are reported in Figure 2.

Polymer and plasticizing oils are recognizable by the peaks at 2920–2852 cm^−1^ (stretching -CH_2_-) and the shoulder at 2953 cm^−1^ (stretching -CH_3_). The -CH_2_- bending is overlapped by the band at 1455 cm^−1^ due to -O-(C=O)-O- group (CaCO_3_). The -CH_3_- bending appears at 1377 cm^−1^. The band at 720 cm^−1^ is due to the amorphous phase -CH_2_- rocking; the analogous in the crystalline phase at 730 cm^−1^ is not evident here. The insulated propylene structural units in EPDM (amorphous phase) absorb at 970 cm^−1^ and at 1158 cm^−1^ (weak bands); here the second band is hidden [34].

Signals at 3643 cm^−1^ (-Si-O-H), 1092–1088 and 799 cm^−1^ (-Si-O-Si-) are attributed to silicates [35]. Signals at 1795, 1452, 872, and 712 cm^−1^ are attributed to calcium carbonate (calcite).

By comparing the FTIR spectra in Figure 2 it appears that the extraction of plasticizing oils in acetone reduces, as expected, the contribution of the 2920, 2851, and 1377 cm^−1^ bands. The band at 1450 cm^−1^ appears less affected because of the overlapping with the strong absorption of CaCO_3_. Similarly, these absorptions are further reduced by remotion of the sol fraction after SW. Therefore, FTIR accounts for the plasticizing oil extraction in acetone and the polymer sol fraction in swelling (in toluene).

#### 3.1.2. Py-GC/MS

The pyrogram of the extracted EPDM RM is displayed in Figure 3. This is a typical thermogram for EPDM rubber, which thermally degrades by intermolecular transfer forming a series of triplets dominated by α olefins of various lengths with different degrees of methyl branches [36].

PY-GC/MS highlights the nature of the diene in EPDM through several specific degradation products [37]. In this case the peaks at r.t. 3.39 min and 3.45 min correspond to 4-ethylidene-1-cyclopentene and to 3-ethylidene-1-cyclopentene respectively, arising from thermal degradation of 5-ethylidene-2-norbornene (EBN) EPDM units not involved in the crosslinking according to Figure 2.

Several benzothiazole derivatives appear in the pyrogram such as benzothiazole (BT, r.t 11.178 min), methylbenzothiazole, (MeBT, r.t 12.179 min) and 2-(methyl mercapto) benzothiazole (MMBT r.t 16.182 min), indicating that the accelerator in the vulcanization package was benzothiazole-based. All MS spectra are reported in the Appendix A. According to the classical mechanism, BT, alkyl MBT, and polysulfide alkyl BT are formed in the vulcanization of EBN-EPDM [38] however these compounds, being soluble in acetone, are removed in the extracted samples. It follows that their presence in the pyrogram must be ascribed to pyrolysis. Figure 3 describes how BT arises from ENB monosulfide residue. Accordingly, MMBT could arise from EBN polysulfide residues.

In Figure 4 the molecular ion extracted pyrograms of RM and of a devulcanized sample (test 5) are reported either after-acetone extraction or after-SW. It appears that the amount of ethylidene-1-cyclopentenes (*m*/*z* 94) in comparison to BT (*m*/*z* 135) or MeBT (*m*/*z* 149) is higher in the extracted than in the after-SW series. This supports the idea that unvulcanized EBN units are mostly confined to the sol fraction and the monosulfide crosslinkings in the gel phase.

In addition, MMBT (*m*/*z* 181) is only found in the RMs and not in devulcanized samples. This support the hypothesis that polysulfides were destroyed in the devulcanization process.

### 3.2. Quantitative Samples Composition

#### TGA

The composition of the original raw material evaluated from TGA is reported in Figure 5.

After devulcanization the amount of the inorganic fraction (CaCO_3_ and residue) slightly decreases mainly because of the decreasing CaCO_3_ content. On the contrary the organic fraction slightly increases mainly due to increased content of rubber, especially in the tests carried out with screw 1 and 2 (test 2–17). It appears that devulcanization affects the amount of CaCO_3_ (from 15.41% in RM to 14–14.6% in devulcanized samples) possibly by its thermal decomposition and on the amount of rubber (from 32.20% in RM up to 33.19% in devulcanized samples) because the volatilization of plasticizing oils in the devulcanizing equipment (Figure 6).

### 3.3. Determination of the Sample Glass Transition Temperatures

#### DSC

The Tg values of original, after-extraction, and after-SW EPDM series are reported in Figure 7. The higher the Tg, the more rigid is the polymer network. Hence, devulcanized EPDMs are expected to show a lower Tg compared to that of undevulcanized ones. However, the Tg values of the original samples are nearly constant, of RM 213.6 K and those of devulcanized samples spanning from 212.2–215.3 K. This obviously depends on the presence of the plasticizing oils. As seen by TGA data, devulcanized original samples exhibit a slightly lower amount of plasticizer, especially those devulcanized with screw 1 and 2 (tests 2–17), and consequently they have a slightly higher Tg. 

On the contrary, when the plasticizer has been removed from the samples as in the extracted series, the Tg is higher in comparison to the corresponding original samples. In addition, in this series RM has a higher Tg (229.9 K) compared to those of the devulcanized samples (220–226 K). This supports the loss of connecting ties in the network during devulcanization. Eventually the EPDM samples after-SW exhibit an even large Tg compared to those of the other series, and this supports the view that a (mobile) sol fraction is removed from the network during swelling. However, RM exhibits similar Tgs either after extraction (229.9 K) or after-SW (231.0 K) meaning that in this case the sol fraction is very low while it increases in the devulcanized samples due to the action of mechanical and thermal forces.

### 3.4. Elemental Analysis

The results of elemental analyses for extracted and after-SW series of EPDM are reported in Table 2.

Other elements come from the inorganic fillers (CaCO_3_ and silicates). Their amount is in line with the inorganic fraction in the original EPDMs, subtracting the plasticizing oils contribution when calculating the percentages of the various components.

C and H arise from the polymer. In addition, part of the C is due to the presence of CaCO_3_. From data in Table 2 it can be seen that:The content of C and H in extracted samples is higher than that in after-SW samples. The polymer is insoluble in acetone (extraction) but soluble in toluene (swelling test). Therefore, these data clearly highlight that during swelling the sol fraction initially present in the network or formed in the devulcanization process is removed from the sample, decreasing the percentage of the polymer in the sample and consequently the % of C and H;In the after-SW series the content of C and H of RM is higher than that of the devulcanized samples, confirming that in the devulcanization process the sol fraction increases;In the extracted series the content of C and H of RM is slightly higher than that of the devulcanized samples. This points contextually with the slight decrease of the other elements percentages, and agrees with the partial decomposition of CaCO_3_ during the devulcanization process.

N and S arise from the vulcanization package based on accelerators (benzothiazole derivatives, as ascertained by Py-GC/MS) and elemental sulfur, used in the vulcanization process. Even if their amount is close to the experimental error several tendencies can be envisaged:The amount of N is similar and remains nearly constant along both the series. It means that the benzothiazole-containing units are neither soluble in acetone nor in toluene. Consequently, it can be argued that these groups are mainly connected, as terminal units, to the crosslinked network, as expected from the vulcanization mechanism;In the after-SW series the amount of S increases in comparison to that in the extracted ones. Considering that the elemental sulfur possibly formed in the devulcanization process is nearly insoluble in toluene, its increase can be due to the remotion of the sol phase, which decreases the amount of polymer in the sample under analysis;In the extracted series the N/S mol/mol ratio is lower for RM (1.20) than in devulcanized EPDM. Considering the insolubility of elemental sulfur and of benzothiazole-units in acetone, this means that possibly part of the elemental S can be removed in the devulcanization process.

### 3.5. Swelling Test: % Devulcanization (%DVZ) and %sol

Results of the swelling tests in term of *%DVZ* and *%sol* are reported in Table 3, Table 4 and Table 5, split up based on the different thermal profiles used in devulcanization. *%random* derived from the Horikx (Equation (10) approach is reported also, which will be discussed later.

There is clear evidence that the higher the thermal profile, the higher *%DVZ* and that when using low and very low rpm the *%DVZ* slightly decreases in comparison to the test carried out at high and medium rpm. As for the screw profiles, comparing the screw profiles 1 and 2, the *%DVZ* obtained with screw profile 2 (characterized by a different number of kneading elements), is slightly lower. The screw profile 3 gives *%DVZ* quite similar to that of screw profile 1 but, if the amount of injected water is high enough (15%) the *%DVZ* slightly decreases, probably due to the temperature drop of the rubber inside the extruder.

The extruder provides mechanical energy to the rubber through correct construction of the screw profile and the screw speed, while the thermal energy is supplied through the extruder heating apparatus. As can be seen from Table 3, Table 4 and Table 5, for this type of rubber, *%DVZ* is always greater when the thermal profile is high and in this case, the influence of the screw speed is less important. The influence of the screw speed, however, is greater at medium/low thermal profiles.

Similarly, the higher the thermal profile, the higher is *%sol*; likewise, at lower rpm, generally, the *%sol* is lower. However, when rpm are very low, *%sol* seems to increase in comparison to low rpm tests. This can be due to the longer residence time that samples experience in these conditions. Again, *%sol* obtained with screw profile n°2, characterized by a different number of kneading elements, seems to be less affected by the rpm, at least at higher thermal profiles. In tests with screw profile n°3, if the amount of injected water is high enough (15%), mostly, *%sol* slightly decreases at medium, low and very low rpm.

### 3.6. The Horikx Analysis

In Figure 8 the Horikx plot for all samples is depicted, where M was supposed as 200.000 a.m.u. Accordingly, with Msub RM 4425 (from swelling test), γi was 45.20. 

All experimental sol fractions values, S*_f_ (=%sol/100)* lay in between the crosslinking model and the random scission model curves; this means that both crosslink scission and random scission occur. However, it should be noted that all the experimental points are close to the crosslink scission model curve and this is indication of a preferential crosslink-scission mechanism during the devulcanization process.

To investigate the relevance of main chain scissions and crosslinking severance in each sample the *%random* was calculated through Equation (10) (Table 3, Table 4 and Table 5). *%random* represents the percentage of experimental sol exceeding that due to the only crosslinking scissions (according to the Horiks model, severage of crosslinkings): against the *%sol* expected from pure random scission (according to the random Horikx model, random scissions), at the same level of devulcanization.

Figure 9 summarizes the results obtained in terms of *%sol* and *%random* for all samples.

There is no global correlation between the *%sol* and the *%random* since their relative trends are interdependent with the process parameters (screw profile, screw speed, thermal profile, and water injection) and this does not allow a general trend to be provided.

However, it should be noted that in general, for the screw profile n°1, the trend of the *%sol* and the *%random* for the tests carried out with a high thermal profile (tests 2–4) is opposite to that obtained at medium (tests 5–7) and low thermal profiles (tests 8, 9) at different screw speeds. For instance, at high thermal profile *%sol* and *%random* decrease as rpm decreases except in test 10, carried out at very low rpm, in which both show a remarkable rise.

It is therefore to be observed that at high temperatures there is a greater selectivity in the breaking of the crosslinks and therefore a lower *%random* by reducing the effect of mechanical shear given by the reduction in rpm. Conversely, at medium temperature, it is mainly the residence time of the material in the extruder, which decreases as rpm increases, that generates a lower *%random* at higher rpm.

*%random* and *%sol* in tests performed with screw profile n°2 are less affected by the process conditions (thermal profile and screw revolution) in comparison to tests performed with screw profile n°1.

For screw profile n°3, there is not a global correlation between the *%random* and water injection at different percentages. However, it should be noted that an increase in the *%random* occur for the tests carried out at high thermal profile and high rpm. On the other hand, it should be noted that in particular for the tests carried out at medium thermal profile and medium rpm with 15% of water (test 26) the *%random* is the lowest for this screw profile.

## 4. Conclusions

Devulcanization represents the ideal recycling process for a homogenous rubber waste stream because it allows revulcanization of devulcanized samples, enabling virtually endless life of the rubber in accordance with the principles of circular economy. Among several devulcanization processes, the thermomechanical devulcanization—using a co-rotating twin-screw extruder—is the more appealing because it is a continuous process, easy to be industrialized, and furthermore the patented Maris process does not require the use of solvents/devulcanizing agents, making it more environmentally sustainable. However, if thermomechanical devulcanization is carried out in not-optimal conditions a still too crosslinked rubber or a too devulcanized rubber is obtained and in both of these cases an unsuccessful revulcanization is expected.

A comprehensive set of analyses were carried out on a post-industrial EPDM seal from the porthole of a washing machine (a typical example of a homogeneous waste stream) devulcanized in a co-rotating twin-screw extruder using different screw profiles, different thermal profiles, different screw speeds, and water injection.

Characterization of devulcanized samples showed that during the devulcanization process, no matter the condition adopted, a small part of the CaCO_3_ was decomposed and a small part of the plasticizing oils volatized (TGA and elemental analyses). The original EPDM was an ENB-EPDM, vulcanized with a benzothiazole vulcanizing package. Benzothiazole residues remained in the network as end groups connected to the rubber by monosulfide and polysulfide bridges. There is some evidence that polysulfide bridges are converted to S plus monosulfide bridges during devulcanization (Py-GC/MS).

Moreover, during devulcanization part of the rubber network is converted to a toluene-soluble sol phase (FTIR and swelling test).

Devulcanization was ascertained by the decreasing Tg of the network after devulcanization (DSC) and, quantitatively, by the results of the swelling test according to the Flory–Rehner theory. Eventually the occurrence of random scission in the rubber chains, which is mostly an undesired phenomenon because it decreases the mechanical performance and the economic value of the rubber, was put to test by comparing the sol fraction experimentally obtained with the expected in the case of only-random or only-crosslinking scissions, according to the Horikx analyses. In all investigated cases the devulcanized samples lay between the two theoretical Horikx models, showing that both random scissions and severance of crosslinking occurred. However, it should be noted that all the experimental points are close to the crosslink scission model curve, and this is an index of the preferential crosslink scission mechanism during the devulcanization process.

As for the devulcanization condition effects, it can be argued that the higher the thermal profile and the higher the rpm, the higher is the percentage of devulcanization. However, the quality of the devulcanized sample in terms of sol fraction and percentage of random scissions depends on the process parameters and the screw profile which affect the rheological properties and the residence time of the rubber in the extruder. Therefore, it is difficult to provide a general behavior for all the performed tests, because the different process parameters affect the process in different ways.

Eventually the screw profile concurs to the efficiency of the devulcanization. The different number of kneading elements along the screw and more in general the screw profile composition affects the percentage of devulcanization, making in some tests the results more dependent on the screw speed. The water injection along the screw affects devulcanization by decreasing the rubber temperature. The water injection also affects the *%sol* and *%random* obtained.

This work enabled the evaluation of the characteristics of the devulcanized material obtained from the devulcanization process in a co-rotating twin-screw extruder, patented by Maris. This evaluation is only the first step related to the concept of circular economy and was useful to obtain information on the % of devulcanization and on the quality of the devulcanized material in terms of *%sol* and *%random*.

The final goal is, therefore, to obtain a new vulcanized compound produced from virgin material and devulcanized material in the highest possible percentage, so as to guarantee chemical–physical properties similar to virgin material, with the aim of replacing it for the same application which allows the reuse of waste material within the production chain. For this reason, there will be significant advantages in terms of environmental impact and cost effectiveness of the entire production process.

However, it must be underlined that the devulcanized product could also be used as such (not vulcanized or vulcanized) or used in very high percentages with the virgin material, in all those applications that do not require specific chemical–physical properties.

## Data Availability

All experimental data are available at University of Turin, M.P. Luda.

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
