# Peer review of "Recycling of EPDM via Continuous Thermo-Mechanical Devulcanization with Co-Rotating Twin-Screw Extruder"

_polymers, 2022, doi:10.3390/polym14224853_

Round 1

Reviewer 1 Report

The whole research is significant and well organized. It is worthy of publication in this Journal.

My only concern is what is reactive temperature? How do you keep the stable extrusion because of solid to liquid transformation during devulcanization.

Reviewer 2 Report

Brunella et al present a very interesting study on recycling. I enjoyed reading this very nice technical work. Some minor comments/suggestions below.

Reclamation and devulcanization: please mention the reactions in a scheme

L 90 Please mention the concept of reactive extrusion with the enhanced mixing. Please refer to React. Chem. Eng. 2022 7, 245

L 100 ref again needed

L 104. Please mention that the 3D positioning of the bonds matters in the efficiency of the break-up. Notable is recent work on such positioning via Monte Carlo. Structure property relationships can be established, also accounting for structural defects. Please refer to Nat. Mater. 2021 20, 1422.

Flory e Rehner => Flory – Rehner; the contribute => the contribution

L 249 please also add the term dangling chains (if opening crosslink). Similar mechanisms have been reported for EVA degradation (Progr. Photovoltaics 2018, 26, 981)
